# Mitigating Forgetting in Online Continual Learning with Neuron Calibration

**Haiyan Yin, Peng Yang, Ping Li**
Cognitive Computing Lab
Baidu Research
10900 NE 8th St. Bellevue, WA 98004, USA
{haiyanyin, pengyang01, liping11}@baidu.com

## Abstract

Inspired by human intelligence, the research on online continual learning aims to push the limits of the machine learning models to constantly learn from sequentially encountered tasks, with the data from each task being observed in an online fashion. Though recent studies have achieved remarkable progress in improving the online continual learning performance empowered by the deep neural networks-based models, many of today's approaches still suffer a lot from catastrophic forgetting, a persistent challenge for continual learning. In this paper, we present a novel method which attempts to mitigate catastrophic forgetting in online continual learning from a new perspective, i.e., neuron calibration. In particular, we model the neurons in the deep neural networks-based models as calibrated units under a general formulation. Then we formalize a learning framework to effectively train the calibrated model, where neuron calibration could give ubiquitous benefit to balance the stability and plasticity of online continual learning algorithms through influencing both their forward inference path and backward optimization path. Our proposed formulation for neuron calibration is lightweight and applicable to general feed-forward neural networks-based models. We perform extensive experiments to evaluate our method on four benchmark continual learning datasets. The results show that neuron calibration plays a vital role in improving online continual learning performance and our method could substantially improve the state-of-the-art performance on all the evaluated datasets.

## 1  Introduction

While humans and animals exhibit remarkable ability to deal with new tasks by effectively adapting their acquired knowledge without forgetting the previously learned skills, in stark contrast, it is challenging for artificial learning systems to effectively deal with continuously inquiring tasks. When switching to a new task from some previously learned ones, there is sometimes a significant drop in performance, where such a phenomenon is also referred to as *catastrophic forgetting*. To overcome such an issue, the research on continual learning has emerged, which defines the learning protocol for the scenarios when the training data of different tasks procedurally arrives in a sequential order [20, 23, 28, 35]. The *catastrophic forgetting* problem is rooted in the model training process, as the gradients to train the model parameters encode great amount of information about the dynamically shifting data distribution seen by the model, which could keep changing throughout the entire continual learning period.

In this paper, we consider a challenging task which poses a more restrictive requirement on the conventional continual learning setting, i.e., *online continual learning* [1]. Inspired by many real life scenarios, *online continual learning* requires the data for each training task to appear as a (one-pass) stream of samples. To tackle such challenges, we focus on the replay-based approach [19, 34], which

35th Conference on Neural Information Processing Systems (NeurIPS 2021).

grants the model with limited access to the data from past tasks for the rehearsal of past experience. To mitigate the *catastrophic forgetting* issue, such methods leverage the replay memory to consolidate the past knowledge in various forms, such as raw data [1] and past gradients [27, 32], to facilitate effective transfer of knowledge. Though the research community has also started to pay attention on a number of non replay-based directions to tackle the continual learning problems, such as leveraging energy-based models [14] and neuron modulation approaches [15], the replay-based methods still play a pivot role in continual learning with state-of-the-art performance in many challenging scenarios. However, despite of the promise in their performance, the replay-based attempts easily lead to a data-imbalance issue, which is also formally referred to as the *stability-plasticity dilemma* [21]. In particular, high *plasticity* refers to the case when old experience is drastically forgotten, and high *stability* refers to the case that too much attention has been paid on stabilizing the previously learned knowledge so that the learning hinders the acquisition of knowledge on the new tasks. Lose of balance between *plasticity* and *stability* would deteriorate the performance of continual learning.

The existing replay-based approaches have taken into account of different perspectives of the model training process to remedy the *stability-plasticity dilemma*, such as regularizing the parameter change during training [4, 12, 23], selective memory storage or replay [1, 11], Bayesian and variational Bayesian training [4, 12, 22], and task-specific parameterization of the model [25, 36]. In this paper, we tackle the problem from a novel angle that is distinct to all the aforementioned attempts, i.e., seeking a better balance between *stability* and *plasticity* with neuron calibration. Specifically, we refer to neuron calibration as a process of mathematically adjusting the transformation functions in various layers of deep neural networks. Considering that besides the task setting, *catastrophic forgetting* in contemporary deep learning-based continual learning models is also closely related to the vulnerability of deep neural networks, our proposed neuron calibration approach aims to regularize the parameter update against *catastrophic forgetting* via posing a trainable soft mask on the parameters, which then influences both the model inference process and the model training process through the forward inference path and the backward optimization path. Our work is inspired by the earlier works that seek optimal model generalization through calibrating the neural networks parameters or labels [9, 10], as well as a recent continual learning work that learns task-specific calibrations for retaining task-specific parameters without memory rehearsal [36]. Compared to those works, the neuron calibration approach we propose shares an enjoyable property of being task-agnostic in terms of parameter sharing among the tasks. That is, instead of reserving task-specific parameters for preserving the task knowledge against forgetting, we train a shared calibrated model where we interleave data from different task distributions to effectively optimize the model.

The contributions of our work are three-fold: (i) we introduce a general and light-weight neuron calibration approach to tackle task-incremental online continual learning problems where the models are formulated as feed-forward deep neural networks-based function approximations; (ii) we formulate a novel task-incremental learning paradigm to train the calibrated model with an interleaved optimization scheme to achieve a better balance between *stability* and *plasticity*; (iii) we show through extensive empirical experiments that our proposed method could outperform the state-of-the-art continual learning algorithms as well as the related layer calibration approach with significant margins on all the evaluation datasets.

## 2  Related Work

Based on how the existing methods deal with catastrophic forgetting, following [7, 8, 13, 25], we classify the continual learning methods into three major categories: (i) memory-based, (ii) regularization-based and (iii) dynamic architectural strategies.

**Memory rehearsal** based methods partially stores the data from past tasks into episodic memories to be later used in a form of knowledge rehearsal [29]. The old data are often used in a standard way of performing experience replay [1, 3, 30, 37], but there are also a number of other purposes of use, such as leveraging the data to perform representation learning [26] and forming constraints for model optimization [6, 19]. To facilitate more effective memory rehearsal, there are also a number of methods that attempt to improve the memory selection process for continual learning, such as iCaRL [26] which stores the samples close to the center of each class, MIR [1] that selects most interfered samples for memory rehearsal, and HAL [5] that selects the anchor points of past tasks and interleaves them with new tasks for future training. Overall the memory-based approaches address catastrophic forgetting issue from training with past data but their performance could easily

be degraded under a tight memory restriction. Our proposed neuron calibration approach could effectively scale up the performance standard of memory rehearsal-based approaches under varying memory restrictions.

**Regularization** based methods extend the loss function in continual learning with regularization terms to promote selective consolidation of past knowledge stored in the model parameters. Often, an importance measure over the model parameters need to be established. In [12], an *elastic weight consolidation* (EWC) approach is proposed with an importance measure defined as promoting the parameters that have a higher value in terms of the Fisher information matrix. In [2], a *selfless sequential learning* algorithm is proposed which exploits the sparse and de-correlated representations to avoid overlapping of representations. In [33], hard attention is learned on the task embeddings to identify important neurons during gradient propagation. We inherit the idea of EWC to regularize the degree of parameter change during online continual learning , but such regularization in our method is to work with the specific calibrated neural network models, whereas applying the EWC regularizer by itself often leads to much inferior performance on conventional online continual learning tasks.

**Dynamic architectural** methods attempts to address catastrophic forgetting through approximating the training of a separate network per task. In [31, 33, 36, 38] a sub-network is trained for each task and in [16, 39] the architecture of model grows over time during training. Though isolating shared model parameters could for sure mitigate forgetting of past knowledge, the drawbacks are in terms of extensive resource usage and scalability. On the other hand, such methods also fail to utilize the multi-task nature of continual learning to exploit the relationship between related tasks for achieving better generalization. Our work is mostly related to [25], which engages task-specific embeddings to calibrate the feature output of neural networks and train the task-specific parameters with hold-out validation data to promote generalization. Our method differs from [25] in terms of the following three points: (i) our method is built upon a neuron calibration approach, where such contribution is orthogonal to that from all the previous works; (ii) our proposed method does not engage any task-specific part; (iii) we do not use any hold-out data from the episodic memories during training, considering that holding out data over the episodic memories which typically come with restrictive storage budgets might not be a desirable practice. Our work is also related to [36] which calibrates neural networks to obtain task-specific parameters. However, the method only trains the task-specific calibrator parameters after the first task and freezes the base parameters thereafter. Such model design makes the model to face a potential risk that when the data distributions of the later tasks are significantly biased from the first task, freezing the base model might lead to inferior performance. Our proposed task-agnostic calibrator shares all the parameters among tasks and it trains all its parameters with interleaved optimization throughout continual learning so that it can better exploit task relationships.

## 3 NCCL: (N)euron (C)alibration for online (C)ontinual (L)earning

**Notation.** In this section, we first introduce the notations for online continual learning. Formally, we denote the sequence of training tasks in online continual learning as $\{\mathcal{T}_1, \cdots, \mathcal{T}_T\}$, where $\mathcal{T}_t$ is the $t$-th task. The tasks arrives in order and the training data for each task is observed in an online fashion. Inheriting a memory-based setting, each task is granted with a small amount of storage to save the past data, which is termed as a *episodic memory*. For the task $\mathcal{T}_t$, we denote its training dataset as $\mathcal{D}_t$ and its episodic memory as $\mathcal{M}_t$. The assembly of all the episodic memories for the past tasks prior to $t$ is denoted as $\mathcal{M}_{<t}$. Note that $\mathcal{M}_t$ is a subset sampled from $\mathcal{D}_t$ and in our work we adopt uniform sampling throughout the learning. The objective of the task is to learn a neural networks-based classifier. Our work tackles the image classification-based continual learning problems and thus the classifier is typically modeled as a feed-forward neural network with $L$ layers (i.e., $\{l_i\}_{i=1}^L$), with its corresponding parameters denoted as $\{\theta_i\}_{i=1}^L$.

### 3.1 Neuron Calibration

We introduce a general neuron calibration mechanism to tackle the online continue learning problems on image classification, where the models are formulated as feed-forward neural networks. By applying neuron calibration, we aim to adapt the transformation functions in the deep neural network layers, which aims to effectively mitigate catastrophic change on the model parameters while accomplishing a stable consolidation of knowledge from different tasks. Specifically, in this work,

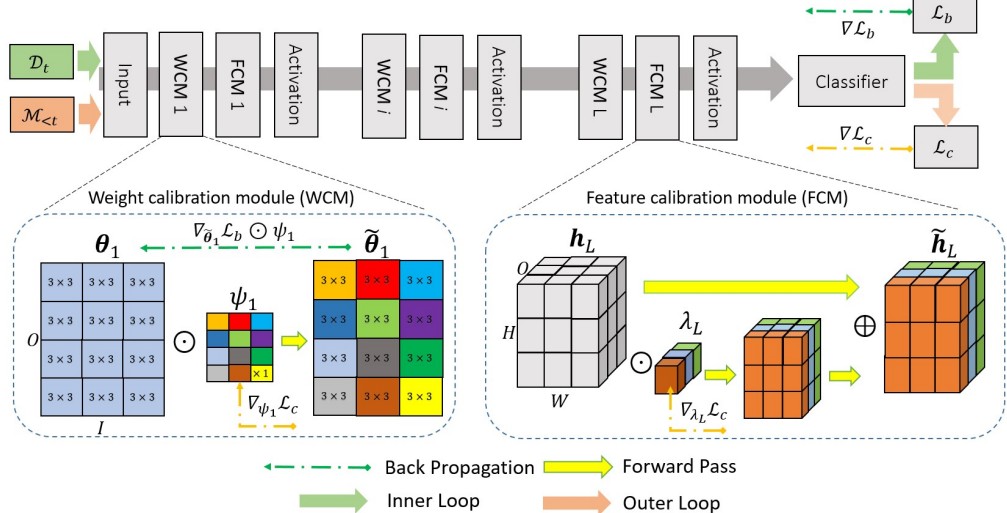

Figure 1: Overview of our proposed Neural Calibration for online Continual Learning (NCCL) framework. NCCL consists of two types of calibration modules: weight calibration module (WCM) and feature calibration module (FCM), which are sequentially applied to the layers in the base model (as shown in the figure) to calibrate the model weights and feature maps, respectively.

we formalize the calibration of two types of commonly adopted transformation layers in feed-forward deep neural networks: *fully connected* layers and *convolutional* layers[1]. Figure 1 provides an illustrative example of our neuron calibration process.

The neural networks model before calibration is denoted as the *base* network. Formally, we introduce two types of general calibration modules to be applied on the *base* network layers: (i) weight calibration module (WCM) and (ii) feature calibration module (FCM). The weight calibration module learns to scale the weights of the parameters from the transformation function whereas the feature calibration module learns to scale the output feature maps predicted by the transformation function [24]. When calibrating the $i$-th layer of a neural network, we use $\theta_i$ and $\widetilde{\theta}_i$ to denote its transformation function parameters before and after applying weight calibration (WCM), and use $\mathbf{h}_i$ and $\widetilde{\mathbf{h}}_i$ to denote the output feature maps before and after applying feature calibration (FCM), respectively.

We first introduce the formulation for WCM. Let $\Omega_{\psi_i}(\cdot)$ denote the weight calibration function employed by the $i$-th layer of the network, which is parameterized by $\psi_i$. Overall, the weight calibration unit is modularized by an element-wise multiplication operation, which is applied between the base network parameters and the calibrator parameters. Specifically, we refer to the $i$-th layer parameters from the *base* network transformation function as $\theta_i$. Then the weight calibration function is defined in the following manner,

$$
\Omega_{\psi_i}(\theta_i) = \begin{cases} \text{tile}(\psi_i) \odot \theta_i, & \psi_i \in \mathbb{R}^{O \times I} & \text{(Convolution Layer)}, \\ \text{tile}(\psi_i) \odot \theta_i, & \psi_i \in \mathbb{R}^{O} & \text{(Fully Connected Layer)}, \end{cases} \tag{1}
$$

where $\odot$ denotes element-wise multiplication, $O$ and $I$ denote the number of output channels and input channels, respectively. To reduce the size of the calibrator parameters for efficient computation, we specify the calibration size to be much smaller than the original size of $\theta_i$. To this end, the calibration for the *convolutional* layers and *fully connected* layers is specified to work at per-channel-level and per-feature-level, respectively. To scale up the shape of the calibrator parameters $\psi_i$ to match that for $\theta_i$, a tile($\cdot$) function is applied on the calibrator parameters $\psi_i$. With the aforementioned weight calibration approach, the calibrator module plays a crucial role during the model training process: at the forward inference path, it scales the value of the *base* network parameters to make prediction; at the backward optimization path, it serves as a prioritized weight to regularize the update

---

[1]Calibrating the fully connected and convolutional layers suffices to deal with primary continual learning tasks on image classification, and our method could also be easily extended to calibrate other types of layers.

on important parameters (e.g., $\nabla_{\theta_1}\mathcal{L}_b$ is derived in the form of $\nabla_{\tilde{\theta}_1}\mathcal{L}_b \odot \text{tile}(\psi_1)$, scaled by the calibrator parameters).

By applying weight calibration on neural network layers, the transformation function at the $i$-th layer $\mathcal{F}_{\tilde{\theta}_i}$ is parameterized by $\tilde{\theta}_i$ and produces the output as follows,

$$h_i = \mathcal{F}_{\tilde{\theta}_i}(\tilde{h}_{i-1}) \quad \text{s.t.} \quad \tilde{\theta}_i = \Omega_{\psi_i}(\theta_i). \tag{2}$$

The output $h_i$ of the transformation function is then processed by a feature transformation module (FCM) to generate the calibrated feature map for that layer. We use $\Omega_{\lambda_i}(\cdot)$ to denote the feature transformation function at the $i$-th layer, parameterized by $\lambda_i$. With FCM, the calibration parameters also interact with the feature map $h_i$ with a multiplicative operation. Specifically, the calibrated feature map is computed as:

$$\Omega_{\lambda_i}(h_i) = \begin{cases} \text{tile}(\lambda_i) \odot h_i, & \psi_i \in \mathbb{R}^O \text{ and } h_i \in \mathbb{R}^{H \times W \times O} \quad \text{(Convolution Layer)}, \\ \lambda_i \odot h_i, & \psi_i \in \mathbb{R}^O \text{ and } h_i \in \mathbb{R}^O \qquad \text{(Fully Connected Layer)}, \end{cases} \tag{3}$$

where $O$ is the output (channel) size for *fully connected* layers and the *convolutional* layers.

In the end, the outputs from (2) and (3) get added up in an element-wise manner by a residual connection. This is followed by normalization and activation operations to produce a final output for that layer. In summary, the overall calibration process for the $i$-th layer could be formulated as follows,

$$\tilde{h}_i = \sigma\left(\mathcal{BN}\left(\Omega_{\lambda_i}\left(\mathcal{F}_{\tilde{\theta}_i}\left(\tilde{h}_{i-1}\right)\right) \oplus \mathcal{F}_{\tilde{\theta}_i}\left(\tilde{h}_{i-1}\right)\right)\right), \quad \text{s.t.} \quad \tilde{\theta}_i = \Omega_{\psi_i}(\theta_i) \tag{4}$$

where $\mathcal{BN}(\cdot)$ denotes the batch normalization, $\oplus$ denotes an element-wise addition operator, and $\sigma(\cdot)$ is an activation function. $\tilde{h}_i$ is sent as input to the $i+1$-th layer in the feed-forward neural network. All the calibrator parameters are initialized with a value of 1 at the start of training. We demonstrate an example case of applying neuron calibration on a CNN-based model in Figure 1.

## 3.2 Learning Calibration Parameters

We propose a new learning paradigm to train the calibrated neural networks model. During model training, we aim to utilize the training of the calibrator parameters to maximally prevent the catastrophic forgetting issue for online continual learning. Considering that such *plasticity* in training is often reflected as drastic changes in parameter values, we set the objective for this calibrator learning as to regularize the parameter change after consolidating the new knowledge and past knowledge (from $\mathcal{D}_t$ and $\mathcal{M}_{<t}$) not to be biased too much from the parameter values before the update. Given the formulation of neuron calibration, the parameter change to be regularized is in terms of the *calibrated* values rather than that for the vanilla base model parameters.

To formulate the loss function for the training of calibrator parameters, we inherit the *elastic weight consolidation* (EWC) approach proposed in [12] . Specifically, EWC approximates the true posterior distribution for the continual learning parameters by a Gaussian distribution given by the mean from the previous tasks and a diagonal precision given by the Fisher information matrix. In our work, we formulate a weight consolidation process to avoid catastrophic change on *calibrated* parameters. In particular, the consolidation process takes place after the *base* model parameters are trained to absorb new knowledge and rehearsing the past knowledge by replaying data from the episodic memory. Then we train the calibrator parameters with the following loss function,

$$\mathcal{L}_c(\{\psi, \lambda, \theta\}, (\mathbf{x}, y, k)) = \underbrace{\frac{1}{2}\text{vec}\left(\tilde{\theta} - \tilde{\theta}^t\right)^\top \Lambda_t \text{vec}\left(\tilde{\theta} - \tilde{\theta}^t\right)}_{\text{term (a)}} + \underbrace{\beta D_{KL}\left(S(\frac{\hat{z}}{\tau})\middle\| S(\frac{\hat{z}_k}{\tau})\right)}_{\text{term (b)}}, \tag{5}$$

where $\beta$ is a trade-off parameter, $S(\cdot)$ is the softmax function, $\tau$ is a softmax temperature, $\hat{z}$ are the logits of current model's prediction before the softmax function, and $\hat{z}_k$ is the logits predicted by previous models saved to the episodic memory. The operator $vec\ (\cdot)$ stacks the tensor into a vector.

The matrix $\Lambda_t$ in term (a) are the Fisher information matrix, which is obtained from a knowledge distillation loss for memory rehearsal. To consolidate the weight in terms of the *calibrated* values,

**Algorithm 1:** Neural Calibration for online Continual Learning (NCCL) Algorithm

---

**Input:** Base model $f_\theta$, calibrator $f_{(\psi,\lambda)}$ learning rate $\alpha_{in}/\alpha_{out}$, outer loss weight $\beta$, training dataset $\{\mathcal{D}_1^{tr}, ..., \mathcal{D}_T^{tr}\}$, testing dataset $\{\mathcal{D}_1^{te}, ..., \mathcal{D}_T^{te}\}$, episodic memory $\{\mathcal{M}_1, ...\mathcal{M}_T\}$, observation batch size $bs_o$, rehearsal batch size $bs_r$ and a confusion matrix to store online learning results $\mathcal{CM} \in \mathbb{R}^{T \times T}$.

**Output:** Base model $f_\theta$, calibrator $f_{(\psi,\lambda)}$, ACC ($\uparrow$), FM ($\downarrow$) and LA ($\uparrow$).

1 **function** train_and_eval
2      Randomly initialize $\theta$, $\psi$ and $\lambda$.
3      Initialize $\mathcal{M}_t \leftarrow \emptyset$ for $t = \{1, ..., t\}$.
4      **for** $t \leftarrow 1 \, to \, T$ **do**
         /* Training                                                  */
5          **for** $b \leftarrow 1 \, to \, n_{batch}$ **do**
6              Observe a now stream of online data $\{x_i, y_i^t\}_{i=1}^{bs_o}$ from task $t$.
7              Update episodic memory for task $t$: $\mathcal{M}_t \leftarrow (\mathcal{M}_t, \{x_i, y_i^t\}_{i=1}^{bs_o})$.
8              **for** $i \leftarrow 1 \, to \, step$ **do**
                 /* Inner step: consolidate new knowledge and old knowledge     */
9                  Sample $bs_o$ samples from task $t$: $\mathcal{B}_t \leftarrow \mathcal{M}_t$        // intra-class sampling
10                 Sample $bs_r$ samples from old tasks: $\mathcal{B}_{<t} \leftarrow \mathcal{M}_{<t}$    // inter-class sampling
11                 $\mathcal{B}^{inner} \leftarrow \mathcal{B}_t \cup \mathcal{B}_{<t}$
12                 $\theta' \leftarrow \theta - \alpha_{in} \nabla_\theta \mathcal{L}_b(\mathcal{B}^{inner}; \theta, \psi, \lambda)$
                 /* Outer step: elastic weight consolidation                    */
13                 $\mathcal{B}^{outer} \leftarrow \mathcal{M}_{<t}$
14                 $\psi' \leftarrow \psi - \alpha_{outer} \nabla_\psi \mathcal{L}_c(\mathcal{B}^{outer}; \theta', \psi, \lambda)$
15                 $\lambda' \leftarrow \lambda - \alpha_{outer} \nabla_\psi \mathcal{L}_c(\mathcal{B}^{outer}; \theta', \psi, \lambda)$
16              **end**
17          **end**
18          Update logits prediction for data from current task: $\hat{z} \leftarrow f(\mathcal{B}_{i=1}^{n_{batch}}; \theta, \psi, \lambda)$
         /* Evaluation                                                  */
19          **for** $te \leftarrow 1 \, to \, T$ **do**
20              Evaluate testing accuracy ($ACC_{te}^{(t)}$) for the current model on $\mathcal{D}_{t_{eval}}^{te}$.
21              Update $\mathcal{CM} \leftarrow \mathcal{CM} \cup ACC_{te}^{(t)}$.
22          **end**
23      **end**
24      Compute ACC ($\uparrow$), FM ($\downarrow$) and LA ($\uparrow$) from $\mathcal{CM}$.
25      **return** *Base model $f_\theta$, calibrator $f_{(\psi,\lambda)}$, ACC ($\uparrow$), FM ($\downarrow$) and LA ($\uparrow$).*
26 **end**

---

the Fisher information matrix needs to be computed upon the gradients evaluated on the *calibrated* parameter values. Formally, we derive its gradient by the chain rule as follows,

$$\nabla_{\widetilde{\theta}^t} D_{KL}\left(S(\frac{\hat{z}}{\tau}) \middle\| S(\frac{\hat{z}_k}{\tau})\right) = \nabla_{\theta^t} D_{KL}\left(S(\frac{\hat{z}}{\tau}) \middle\| S(\frac{\hat{z}_k}{\tau})\right) \cdot \underbrace{\nabla_{\widetilde{\theta}^t} \theta^t}_{\text{tile}(\psi)^{-1}} \tag{6}$$

Thus the Fisher information matrix distinguishes the parameters that are important to retain past knowledge and penalizes their weight change appropriately following the term (a). We add an additional knowledge distillation term (b) to let the training take care of the stabilization issue as well, where the stabilization issue refers to the case of avoiding the parameters to experience much sudden change which is considered as a potential source of catastrophic forgetting.

### 3.3 Interleaved Optimization

We formulate the optimization process to train the calibrated model under an interleaved optimization schema [17], with the parameters from the base model and those from the calibrator model being opti-

mized by their respective loss functions. Formally, the training process can be formulated as follows:

$$\text{Outer Loop:} \quad (\psi^*, \lambda^*) \quad = \arg\min_{(\phi, \psi)} \mathcal{L}_c((\psi, \lambda), \theta^*, \mathcal{M}_{<t}), \tag{7}$$

$$\text{Inner Loop:} \quad \text{s.t.} \quad \theta^* \quad = \arg\min_{\theta} \mathcal{L}_b((\psi, \lambda), \theta, \mathcal{M}_{\leq t}), \tag{8}$$

where $(\psi^*, \lambda^*)$ denote the optimal calibrator parameters when fixing base model $\theta$, and vice versa. At the inner loop, the base model parameters are updated to learn new knowledge from $\mathcal{D}_t$ and rehearse old knowledge from $\mathcal{M}_{<t}$. Thus the data for the inner loop update comes from $\mathcal{M}_{\leq t}$ with $\mathcal{L}_b$ being a cross-entropy loss. At the outer loop, the calibrator parameters are updated to consolidate the weight update from the inner loop against *catastrophic* change by employing a loss $\mathcal{L}_c$ which is introduced in Section 3.2. During the interleaved optimization process, we first fix $(\psi_t, \lambda_t)$ and take gradient steps with regard to $\theta$ as follows:

$$\theta_{t+1} \leftarrow \theta_t - \alpha_{in} \bigtriangledown_\theta \mathcal{L}_b((\psi_t, \lambda_t), \theta_t, \mathcal{M}_{\leq t}). \tag{9}$$

Then, we go on to optimize the calibrator when the inference takes place with the updated base model,

$$\psi_{t+1} \quad \leftarrow \psi_t - \alpha_{out} \bigtriangledown_\psi \mathcal{L}_c((\psi, \lambda), \theta_{t+1}, \mathcal{M}_{<t}), \tag{10}$$

$$\lambda_{t+1} \quad \leftarrow \lambda_t - \alpha_{out} \bigtriangledown_\lambda \mathcal{L}_c((\psi, \lambda), \theta_{t+1}, \mathcal{M}_{<t}), \tag{11}$$

where $\alpha_{in}$ and $\alpha_{out}$ are the learning rates for the inner loop update and the outer loop update, respectively. By employing the calibrated parameterization of the neural networks-based model and optimizing it with the interleaved learning scheme, our method could potentially tackle the *catastrophic forgetting* issue in online continual learning with greater effectiveness than many conventional approaches. We present the detailed algorithm in Algorithm 1.

## 4 Experiments

In this section, we demonstrate the empirical evaluation results on comparing our method with a number of closely related baselines under various experimental settings. Our method is implemented using the PaddlePaddle (PArallel Distributed Deep LEarning) framework.

### 4.1 Benchmark Datasets

We consider the following four continual learning datasets as our evaluation testbeds: **Permuted MNIST** (denoted as **pMNIST**) [19], **Split CIFAR** [19], **Split miniImageNet** [6], and a continual learning benchmark created with the real world dataset **Split CORe50** [18]. For fair comparison in **Split miniImageNet** and **Split CORe50**, we adopt identical datasets as those used in [25]. For online continual learning with **pMNIST**, we create 20 independent tasks with each task being created by randomly permuting the order of the pixels in an MNIST image. Each task in **pMNIST** consists of 1000 samples for the 10 MNIST classes and all the tasks share a common label space. **Split CIFAR** is constructed by evenly splitting the 100 classes for CIFAR into 20 disjoint subsets, where each task takes 2500 training samples. **Split miniImageNet** consists of 17 tasks, where each task has 5 classes disjoint from other tasks and each task takes 2500 samples. **Split CORe50** is constructed by evenly splitting the 50 classes in the dataset into 10 disjoint tasks, where each task has 5 classes. The details on the dataset statistics is available in appendix.

### 4.2 Architectures

For the pMNIST experiments, we use a multi-layer perceptron with two hidden layers of size 256 [25]. For Split CIFAR and Split miniImageNet experiments, we use a reduced ResNet18 with three times fewer CNN filters than the full ResNet18 [19, 25]. For CORe50 experiments, we use a pretrained ResNet18 where the pretrained parameters are loaded from *paddleCV* in PaddlePaddle (we do not use a new model to be trained from scratch in this task because such models yield extremely poor accuracy which is at around 30% during most of the learning period and thus fail to be a valid domain to compare the continual learning ability for different approaches). We show the model capacity in terms of trainable parameter sizes for different models in Table 1. The results demonstrate that our proposed calibration approach leads to moderate increases in parameter size compared to its corresponding backbone models for all the architectures. The parameter increase is approximately 10% ~15% for the CNN-based models, and that for the multi-layer perceptron-based models are much more smaller, i.e., much less than 1%.

Table 1: Light-weight model complexity w.r.t parameter size for various network architectures. NCCL results in moderate parameter increase over its backbone architectures in all testified datasets.

| Architecture | Backbone # Params | Structure | NCCL # Params | Increase |
|---|---|---|---|---|
| MLP (pMNIST) | 269,322 | Linear | 270,366 | 0.39% |
| ResNet18-reduced (CIFAR) | 1,109,240 | Linear, Convolution | 1,265,440 | 14.08% |
| ResNet18-reduced (miniIMN) | 1,106,825 | Linear, Convolution | 1,263,025 | 14.11% |
| ResNet18-full (CORe50) | 11,202,162 | Linear, Convolution | 12,430,450 | 10.96% |

## 4.3 Baselines and Evaluation Metrics

For evaluation, we compare our method with the following continual learning baselines: **GEM** [19] and **AGEM** [6] which are two representative gradient projection-based rehearsal approaches which find and store the important sub-spaces for past gradients and let the learning on new task to take gradient steps orthogonal to the saved sub-spaces, **MER** [27] which performs experience replay with reptile update that regularize updated parameter values to be close to the initial values, **MIR** [1] which retrieves samples that would be maximally interfered by the new samples to update the model, and **CTN** [25] which is a recently published state-of-the-art method that performs task-specific scaling on the output of the prediction layers. Apart from the established baselines, we also consider an **independent** baseline, where each of the online continual learning task is trained with an independent model that share the same model architecture with each other, an **offline** baseline where the task setting assumes the data for all the tasks are available at hand in an off-line fashion where we do not specify a limit for the epochs we train the model. We denote our proposed method (N)euron (C)alibration for online (C)ontinual (L)earning as **NCCL**. During evaluation, we adopt identical datasets and task ordering for each method. It is also important to note that during the online continual learning process, each *mini*-batch of samples could be used to update the model for several times and each of the considered method shares such permission. Each score reported is derived from 10 independent runs. We present the detailed hyperparameter settings for all the baseline methods as well as NCCL on all the datasets in appendix.

To evaluate the methods, we adopt three standard evaluation metrics on all the datasets: **ACC** ($\uparrow$) (higher is better) [19], **FM** ($\downarrow$) (lower is better) [4], and **LA** ($\uparrow$) (higher is better) [27]:

- **Averaged Accuracy** (**ACC** $\uparrow$): is the continual learning accuracy evaluated after the model has been trained on all the tasks, i.e., ACC $= \frac{1}{T} \sum_{i=1}^{T} a_{i,T}$, where $a_{i,j}$ denotes the accuracy on task $i$ after training on the task $j$.
- **Forgetting Measure** (**FM** $\downarrow$): the average difference between the final performance obtained for each task compared to the best performance on each task, i.e., FM $= \frac{1}{T} \sum_{i=1}^{T} |a_{i,T} - a_i^*|$, where $a_i^*$ denotes the best performance on task $i$.
- **Learning Accuracy** (**LA** $\uparrow$): the average of best accuracy evaluated through continual learning for each task domain, i.e., LA $= \frac{1}{T} \sum_{i=1}^{T} a_{i,i}$.

We present the empirical results on the continual learning benchmark datasets in Table 2. We observe that our method NCCL consistently outperforms the other baseline approaches with large margins and achieves state-of-the-art performance on all the datasets. Even compared with CTN, a strong baseline with remarkable performance records, our method still outperform it on all the evaluation domains with noticeable margin. Overall, compared to CTN, the improvement of our method in terms of ACC ($\uparrow$), FM ($\downarrow$) and LA ($\uparrow$) are 4.67%, 1.62% and 2.79%, which are averaged among the four datasets. On *Split CIFAR*, *Split miniImageNet* and *CORe50*, our method results in (almost) the highest LA ($\uparrow$) which records the best accuracy achieved throughout the continual learning process. We wish to remind that *Independent* and *Offline* are two baselines created by abusing the data accessibility and model consistency to demonstrate the potential upper bound of the online continual learning performance. From the results presented in Table 2, we observe that the performance of our method is close to the *Independent* baseline on the task *pMNIST* and better than it on the task *Split CIFAR* and *Split miniImageNet*. This might due to the reason that our model training could effectively exploit the multi-task property among the tasks. Moreover, in terms of the best accuracy achieved for each task throughout the learning (i.e., LA ($\uparrow$)), our method is better than the strong baseline *Independent* on all the datasets except for *CORe50* where the gap is quite small.

Table 2: Evaluation results on four continual learning benchmark datasets. The results are obtained with a replay memory size of 50. We adopt identical (backbone) network architecture for all the compared methods on all the datasets. Each score is derived from 10 independent runs.

| Method | Split CIFAR | | | Split miniImageNet | | |
|---|---|---|---|---|---|---|
| | ACC (↑) | FM (↓) | LA (↑) | ACC (↑) | FM (↓) | LA (↑) |
| GEM | 62.85 ±0.83 | 9.60±0.76 | 71.91±0.78 | 59.36±1.28 | 6.97±0.93 | 65.35±0.67 |
| AGEM | 60.43±2.35 | 9.60±2.65 | 69.20±0.79 | 52.70±2.12 | 7.52±1.80 | 58.33±1.01 |
| MER | 61.83±1.18 | 8.44±0.94 | 69.03±1.30 | 55.49±2.01 | 9.11±2.40 | 62.33±1.65 |
| MIR | 62.51±1.45 | 8.29±1.12 | 69.20±1.16 | 57.25±1.04 | 5.73±0.79 | 60.23±1.16 |
| CTN | 68.62±0.59 | 5.92±0.73 | 73.83±0.71 | 65.25±1.65 | 3.78±2.31 | 67.54±2.87 |
| NCCL (ours) | **74.39±0.97** | **4.88±0.97** | **78.28±0.53** | **69.49±0.92** | **3.36±0.91** | **70.69±1.06** |
| Independent* | 70.78±0.73 | 0.0 | 70.78±0.73 | 64.42±1.13 | 0.0 | 64.42±1.13 |
| Offline | 76.98±0.43 | – | – | 70.47±0.99 | – | – |

| Method | pMNIST | | | CORe50 | | |
|---|---|---|---|---|---|---|
| | ACC (↑) | FM (↓) | LA (↑) | ACC (↑) | FM (↓) | LA (↑) |
| GEM | 74.57±0.10 | 7.40±0.11 | 79.13±0.42 | 77.48±2.72 | 12.12±4.49 | 87.92±0.37 |
| AGEM | 69.50±0.76 | 13.10±0.63 | 82.61±0.41 | 78.13±2.14 | 11.60±1.62 | 88.30±0.65 |
| MER | 75.75±0.65 | 8.74±0.73 | 83.52±0.38 | 82.75±1.30 | 7.54±0.79 | 88.04±1.21 |
| MIR | 78.31±0.63 | 7.15±0.67 | 84.48±0.28 | 83.34±1.48 | 7.30±1.32 | 89.52±0.45 |
| CTN | 79.70±0.44 | 5.08±0.44 | 84.08±0.36 | 83.87±1.13 | 5.71±1.08 | 88.93±0.53 |
| NCCL (ours) | **83.47±0.43** | **3.44±0.26** | **86.59±0.25** | **88.80±0.32** | **2.22±0.49** | **90.12±0.38** |
| Independent* | 83.50±0.39 | 0.0 | 83.50±0.39 | 90.34 ±0.45 | 0.0 | 90.34±0.45 |
| Offline | 89.05±0.27 | – | – | 90.89 ±0.44 | – | – |

## 4.4 Evaluation Results on Continual Learning Benchmarks

We also evaluate our method under the cases when online continual learning is performed with varying memory size per task. We evaluate the methods when the memory size per task is obtained from the following set: {50, 100, 150, 200, 250}. We train the methods on two tasks, Split CIFAR and Split miniImageNet, and show the evaluation results for NCCL, CTN, MIR and GEM in Figure 2.

The performance standard for AGEM and MER is far from the presented baselines and we show them in appendix. From the results, we find all the evaluated methods result in consistent performance improvement as the memory size increases. Overall, NCCL results in superior performance than the baseline approaches under all of the evaluated settings for memory size. We also notice that the baseline approaches could hardly outperform the Offline baseline under the restrictions of memory size per task, but our method could outperform it at the early stages of the curves. We clarify that it is possible for the method to pass the Offline method as the memory size increases [25]. The reason

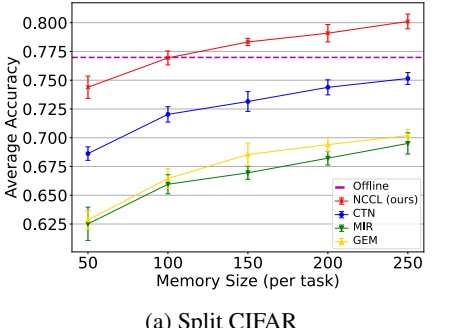

(a) Split CIFAR

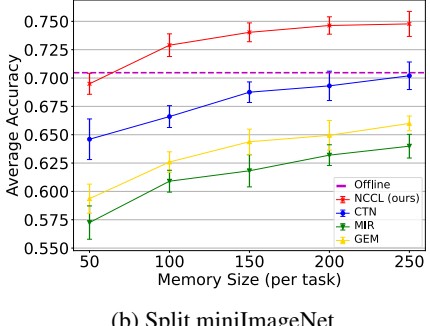

(b) Split miniImageNet

Figure 2: Evaluation results with varying memory size per task on the following two benchmark datasets: (i) Split CIFAR and (ii) Split miniImageNet. The presented values correspond to the measures in terms of ACC (↑). Each score is derived from 10 independent runs.

Table 3: Ablation study results on comparison with Calibrating CNNs for Lifelong Learning (CCLL).

| | Update Calibrator | Update Base | Split CIFAR | | | Split miniImageNet | | |
|---|---|---|---|---|---|---|---|---|
| | | | ACC(↑) | FM (↓) | LA(↑) | ACC(↑) | FM (↓) | LA(↑) |
| CCLL | ✓ | ✗ | 64.77±0.73 | 0.0 | 64.77±0.73 | 60.56±1.26 | 0.0 | 60.56±1.26 |
| | ✓ | ✓ | 74.49±0.79 | 0.0 | 74.49±0.79 | 68.80±1.17 | 0.0 | 68.80±1.17 |
| NCCL | - | - | 74.39±0.97 | 4.88±0.97 | 78.28±0.53 | 69.49±0.92 | 3.36±0.91 | 70.69±1.06 |

that our method outperforms Offline under partial memory capacity settings could possibly due to that NCCL could effectively mitigate catastrophic forgetting while exhibiting a desirable ability to transfer knowledge during its multi-task training. Such findings reveal that the replay-based online continual learning setting could potentially pose an effective curriculum for the learning of multiple tasks and become a desirable choice for multi-task training.

### 4.5 Additional Results on Comparing with CCLL

We mostly consider well adopted memory rehearsal-based algorithms for benchmark comparison. In this section, we extend such comparison and present additional results on comparing with another closely related baseline to ours which calibrates CNN models without memory rehearsal, termed Calibrating CNNs for Lifelong Learning (CCLL) [36].

CCLL calibrates the *feature outputs* of the CNN layers with additional calibrating *layers*, to adapt the model to new tasks. It trains all the model parameters on the first task, and only trains the *calibrator* parameters and the *classifier* parameters on the subsequent tasks with the *base* parameters being fixed. Thus it needs to store the parameters for each task and it requires task identities at the inference time during testing. Moreover, CCLL is sensitive to the choice of the first task, as the base parameters are desired to be trained on informative tasks that are related to later tasks. As such, the motivation for CCLL is essentially different from ours, as we train a task-agnostic calibrator with memory rehearsal and our method calibrator is general to be applied on multiple layers.

We present the comparison results with CCLL in Table 3. We carefully tuned the parameters for CCLL and the settings are available in Section C of appendix. Overall, we noticed that though CCLL alleviates forgetting by storing historical parameters which results in zero FM(↓), the ACC(↑) of CCLL (train calibrator+classifier w/o memory rehearsal) is much inferior than our method NCCL (train all parameters with memory rehearsal). For sanitary checking, we adapt CCLL model to create a variant of CCLL that updates both the calibrator parameters and base parameters. We notice that when updating all the model parameters, the performance is apparently higher than the original CCLL and the setting is essentially identical to the *independent* baseline except their difference in model architecture. The CCLL variant that updates all parameters result in ACC(↑) scores that are close to our method in both evaluation domains. But that updates partial parameters following the original proposed form of CCLL results in much inferior results than NCCL. Overall, the results demonstrate that calibrating on CNN weights with memory rehearsal is a promising direction to consider.

## 5 Conclusion

In this paper, we present a novel online continual learning framework for task-incremental learning problems, which sheds light to a new direction to tackle online continual learning problems with neuron calibration. We formulate a general neuron calibration approach complemented by an interleaved optimization scheme for effective model training. Our proposed solution enjoys a considerable level of generality so that it could potentially tackle many online continual learning problems which employ feed-forward neural networks. One possible limitation for our method is that our work focuses on establishing the generality of the method without considering much on the diversity of neural network layers. Specifically, the formulation we propose tackles the calibration of the feed-forward neural network layers without dealing with other types of architectures, such as the networks with recurrent or recursive structures. Extending our work to develop a more inclusive neuron calibration framework is a promising future research direction to consider. Also, it is worth considering to adapt our method to be applied on another type of more challenging continual learning problems which is under the class-incremental settings.

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
