# Mitigating Forgetting in Online Continual Learning with Neuron Calibration

**Haiyan Yin, Peng Yang, Ping Li**
Cognitive Computing Lab
Baidu Research
10900 NE 8th St. Bellevue, WA 98004, USA
{haiyanyin, pengyang01, liping11}@baidu.com

This **appendix** is organized as follows:

- **Section A**: the detailed dataset statistics and a summary of model properties w.r.t. online continual learning setting.

- **Section B**: additional experiment results on comparing with CTN.

- **Section C**: hyperparameter settings for all the algorithms.

## A   Task Specifications

We present the details on each dataset in Table 4. Under the *online continual setting*, the tasks are observed following a fixed order and the data from each task is observed as a (one-pass) stream of samples. The batch size is 10 for all the datasets. We do not randomize the order of tasks or optimize the task orders. The task order is defined by processing the original datasets (CIFAR, miniIMN and CORe50) or random (pMNIST). We also present a summary of the key model properties with regard to the online continual learning setting in Table 5. The term *store historical params/grads/logits (training)* refers to the case that historical information about the model, such as its parameters, gradients or logit outputs, needs to be accessed for optimizing the model at its training time.

Table 4: Details on the benchmark datasets

| Dataset | #Tasks | # Classes per task | # Training | # Testing | Dimension | Disjoint label space |
|---------|--------|--------------------|------------|-----------|-----------|----------------------|
| pMNIST | 20 | 10 | 1000 | 10,000 | $28 \times 28$ | ✗ |
| CIFAR | 20 | 5 | 50,000 | 10,000 | $3 \times 32 \times 32$ | ✓ |
| miniImageNet | 17 | 5 | 42,500 | 8,500 | $3 \times 84 \times 84$ | ✓ |
| CORe50 | 10 | 5 | 113,894 | 44,971 | $3 \times 224 \times 224$ | ✓ |

Table 5: Details on the model properties w.r.t. the online continual learning setting for each method considered. Note that all the *task*-incremental methods require the access to Task ID during inference.

| Method | Use Episodic Memory | Use Task-Specific Parameters | Need Task ID During Inference | Store Historical Params/Grads/Logits (training) | Store Historical Params (testing) |
|--------|---------------------|------------------------------|-------------------------------|--------------------------------------------------|-----------------------------------|
| Offline | ✗ | ✗ | ✗ | ✗ | ✗ |
| Independent | ✗ | ✓ | ✓ | ✗ | ✓ |
| GEM [19] | ✓ | ✗ | ✓ | ✓ | ✗ |
| AGEM [6] | ✓ | ✗ | ✓ | ✓ | ✗ |
| MIR [1] | ✓ | ✗ | ✓ | ✗ | ✗ |
| MER [27] | ✓ | ✗ | ✓ | ✓ | ✗ |
| CTN [25] | ✓ | ✓ | ✓ | ✓ | ✗ |
| CCLL [36] | ✗ | ✓ | ✓ | ✗ | ✓ |
| NCCL (ours) | ✓ | ✗ | ✓ | ✓ | ✗ |

# B  Additional Results on Comparing with CTN

We present an ablation study to investigate whether task-specific properties could bring positive effect to NCCL. To this end, we consider to employ a light-weight task-specific module proposed in a recent work CTN [25] which has proven to be effective for online continual learning: (1) a light-weight controller module (CTR) which uses a task embedding to generate task-specific calibration over the feature map (2) a validation memory module (VM) which holds out partial samples from the episodic memory to train the controller, for better generalization. We create two variants of CTN that employs CTR or both CTR and VM. The results are shown in Table 6. First, we notice that employing the task-specific controller module does not bring positive performance gain over the original NCCL. While the FM($\downarrow$) for NCCL+CTR is slightly better or comparable to NCCL, the ACC ($\uparrow$) and LA($\uparrow$) are substantially lower. Also, we notice that when employing the VM module, the training with hold-out data leads to inferior performance than only using CTR or the original NCCL. The reason might be that holding out samples to create validation memory would let both the base parameters and the calibrator/context parameters be trained by less samples during memory rehearsal and therefore lead to inferior continual learning performance.

Table 6: Ablation study results on exploring the task-specific properties of NCCL. We employ the controller module (CTR) and the validation memory module (VM) from CTN to our method and evaluate whether the task-specific modules bring any improvement. Each method adopts a memory size of 50.

|  | CTR | VM | Split CIFAR | | | Split miniImageNet | | |
|---|---|---|---|---|---|---|---|---|
|  |  |  | ACC($\uparrow$) | FM ($\downarrow$) | LA($\uparrow$) | ACC($\uparrow$) | FM ($\downarrow$) | LA($\uparrow$) |
| NCCL | ✓ | ✗ | 72.56±0.59 | **3.81±0.40** | 74.62±0.71 | 66.21±0.74 | **3.16±0.62** | 66.74±0.98 |
|  | ✓ | ✓ | 71.32±1.00 | 4.61±0.83 | 74.56±0.80 | 64.47±0.56 | 0.38±0.45 | 66.03±0.49 |
| (ours) | ✗ | ✗ | **74.39±0.97** | 4.88±0.97 | **78.28±0.53** | **69.49±0.92** | 3.36±0.91 | **70.69±1.06** |
| CTN | ✓ | ✓ | 68.62±0.59 | 5.92±0.73 | 73.83±0.71 | 65.25±1.65 | 3.78±2.31 | 67.54±2.87 |

# C  Hyperparameter Specifications

We present the hyperparameter configurations for all the methods considered in our experiment.

- GEM
  - Learning Rate: 0.1 for pMNIST, 0.001 for CIFAR and miniIMN, 0.01 for CORe50
  - Number of gradient updates: 3 (all benchmarks)
  - Margin for QP: 0.5 (all benchmarks)
- AGEM
  - Learning Rate: 0.05 for pMNIST, 0.001 for CIFAR and miniIMN, 0.01 for CORe50
  - Number of gradient updates: 3 (all benchmarks)
  - Margin for QP: 0.5 (all benchmarks)
- MIR
  - Learning Rate: 0.1 for pMNIST and miniIMN, 0.03 for CIFAR and CORe50
  - Replay batch size: 50 (all benchmarks)
  - Number of gradient updates: 3
- MER
  - Learning Rate: 0.1 for pMNIST and miniIMN, 0.03 for CIFAR and CORe50
  - Replay batch size: 64 (all benchmarks)
  - Reptile rate $\beta$: 0.3 (all benchmarks)
  - Number of gradient updates: 3
- CTN
  - Learning Rate: 0.03 for pMNIST, 0.01 (all benchmarks)
  - Number of inner/outer updates: 2 (all benchmarks)
  - Temperature and weight for KL: 5, 100 (all benchmarks)

- – Replay batch size: 64 (all benchmarks)
- – Semantic memory percentage: 20%
- CCLL
  - – Learning Rate: 0.003 (all benchmarks)
  - – Number of gradient updates: 4 (all benchmarks)
- NCCL (ours)
  - – Learning Rate $\alpha_{in}$: 0.1 for pMNIST, 0.01 for CIFAR and miniIMN, 0.003 for CORe50
  - – Learning Rate $\alpha_{out}$: 0.05 for pMNIST, 1e-3 for CIFAR and miniIMN, 1e-4 for CORe50
  - – Replay batch size: 64 (all benchmarks)
  - – Temperature and weight for KL in $\mathcal{L}_b$: 5, 100 (all benchmarks)
  - – Weight $\beta$ in outer loss $\mathcal{L}_c$: 1 (all benchmarks)
  - – Number of gradient updates: 4 (all benchmarks)