# OpenReview forum: "Mitigating Forgetting in Online Continual Learning with  Neuron Calibration"
_NeurIPS.cc/2021/Conference — NeurIPS 2021 Poster_

### Official Review · Reviewer_Ng2E · 2021-07-17

**Rating:** 6
**Confidence:** 5

**Summary:**

This work proposes an online continual learning approach that learns an attention map on the weights (termed as weight calibration module) and another on the features (termed as feature calibration module) of a neural network. The training is done by adding a stability regularization term (similar to CTN) to the original EWC loss and interleaving the optimization of network parameters and attention modules as the new tasks are observed continually. A task-incremental learning scenario is considered, and the accuracy is compared against five other approaches with four benchmark datasets using three different metrics.  With a marginal increase in the number of parameters with the attention maps, we can see considerable improvement in accuracy.

**Limitations And Societal Impact:**

The limitations are not adequately mentioned,  potential negative societal impact of their work are included in the manuscript

**Main Review:**

• Online continual learning is generally associated with memory limitations [1,2,3]. Maintaining a buffer from all the previous tasks can be memory intensive.

•The number of gradients updated for each approach is mentioned in the appendix, but it is not clear if all the data (current batch + memory buffer) is used for multiple updates/steps. Using the current batch multiple times violates the online continual learning scenario

• Is the parameter convergence evaluated after one pass over the data?

• It is claimed that the approach is task-agnostic, but the experiments seem to be in a task-incremental setting (one output head per task) where the task label is provided at training and test time.

• Other related methods [4,5] should be considered in the comparison

• It should be made clear in the introduction on what continual learning scenario is considered [1] and some thoughts on the applicability of this approach from a more realist class-IL scenarios.

• There are also OCL approach without memory buffer that outperform memory-based approaches [3,6]. They should also be included in the related work

[1] Mai, Zheda,  et al. "Online Continual Learning in Image Classification: An Empirical Survey." arXiv preprint arXiv:2101.10423 (2021).
[2] Chen, H.-J., et al. “Mitigating forgetting in online continual learning via instance-aware parameterization” Advances in Neural Information Processing Systems, 33, 2020
[3] Madireddy, Sandeep et al. “Neuromodulated Neural Architectures with Local Error Signals for Memory-Constrained Online Continual Learning”, arXiv preprint arXiv:2007.08159 (2021)
[4] Mirzadeh, Seyed Iman, et al. "Dropout as an implicit gating mechanism for continual learning." In Proceedings of the IEEE/CVF Conference on Computer Vision and Pattern Recognition Workshops, pp. 232-233. 2020.
[5] Yoon, Jaehong, et.al. "Scalable and order-robust continual learning with additive parameter decomposition." arXiv preprint arXiv:1902.09432 (2019).
[6] Li, S., et al. "Energy-Based Models for Continual Learning." arXiv preprint arXiv:2011.12216, 2020


**Time Spent Reviewing:**

4

---

> ### Author Response · Authors · 2021-08-10
> **Responses to Comments from Reviewer Ng2E**
>
> We thank the reviewer for the insightful comments and suggestions to improve our work.
>
> $\textbf{Q1}$: Online continual learning is generally associated with memory limitations [1,2,3]. Maintaining a buffer from all the previous tasks can be memory intensive.
>
> $\textbf{A1}$: The reviewer is right that maintaining a buffer would consume additional memory which might be intensive. For the Split CIFAR and Split MiniImagenet tasks in our paper, the total buffer sizes for them (under the setting of 50  samples per task) are 12.296Mb 167.124Mb respectively. With modern computers, such memory size is rather affordable and with the moderate amount of increase in memory, ER methods could bring prominent performance gain to the performance of (O)CL problems. We would also like to highlight that though our method would consume additional memory for replay buffer but it does not engage additional memory for the model as it uses a unified model to tackle all the tasks. If we wish to achieve promising CL performance without replay buffer, often we need to leverage parameter isolation which leads to heavy memory consumption for storing task-specific parameters (e.g., the *independent* baseline). Compared to parameter isolation, the memory consumption for ER is often more light and while parameter isolation blocks backward transfer effect, ER could allow both forward transfer and backward transfer effect and lead to more desirable performance.
>
>
> $\textbf{Q2}$: It is not clear if all the data(current batch + memory buffer) is used for multiple updates/steps. Using the current batch multiple times violates the online continual learning scenario.
>
> $\textbf{A2}$: We would like to clarify that the current mini batch of data from the ongoing task is used to update the model for multiple time steps. However, this does not violate the online continual learning presumption. The data observation is exactly one-pass stream, i.e., no data is observed repeatedly. For model training , when model observes current batch at time $t$, the data at previous time steps $\{1,2,\cdots,t-1\}$ is unavailable to access (unless you store few points in the buffer. Therefore, learning current batch multiple times does not violet the online setting. Also, note that all the baselines also adopt such multiple time update strategy. So the comparison is valid and fair. For effective learning under the online continual learning regime, such practice is highlighted as one of the most important practices in online continual learning in the survey [1].
>
> $\textbf{Q3}$: Other related methods [4,5] should be considered in the comparison.
>
> $\textbf{A3}$: We thank the reviewer for suggesting the related baselines. We conducted in-depth comparison with [4]. [4] could be considered as a variant of the weight calibration approach where *dropout* is applied over each parameters of the model. We would also like to highlight that [4] does not employ experience replay and the authors attempt to show that *dropout* is effective for CL against forgetting when not using replay, which has been evaluated on two MNIST domains, i.e., permuted MNIST and Rotated MNIST. In the OCL problems that we tackle, adopting [4] without memory rehearsal essentially results in rather inferior performance. In permuted MNIST, the ACC (final accuracy) and FM (forget measure) for [4] is only 44.12 ($\pm$ 2.60) and 38.50 ($\pm$ 2.73), where the FM is 10x of our method NCCL. On Split CIFAR and Split miniImagenet, the performance is even more inferior and we find that the model could not be trained with a large dropout rate as suggested in the paper. Therefore we believe our method is more effective than [4] in general CL tasks. We will add comparison results with [5] in the updated version of the paper.
>
>
> $\textbf{Q4}$: It should be made clear in the introduction on what continual learning scenario is considered [1] and some thoughts on the applicability of this approach from a more realist class-IL scenarios.
>
>
> $\textbf{A4}$: We thank the reviewer for this constructive suggestion. Our work tackles the typical OCL tasks and we will carefully revise the introduction to make this clear cut. The application of our method to a more challenging class-IL setting is considered as a promising future work direction.
>
> $\textbf{Q5}$: There are also OCL approach without memory buffer that outperform memory-based approaches [3,6]. They should also be included in the related work.
>
> $\textbf{A5}$: We thank the reviewer for referring to the two interesting papers that we are not aware of. Both works have novel motivations to tackle the problems (i.e., EBM and neuromodulation) though their empirical results are a bit weak. We would like to extend the discussion we posed in the paper to include those inspiring works however we respectfully stay conservative in terms of the opinion that memory-free works could actually outperform strong replay-based approaches in general OCL problems.
>
>
> $\textbf{Q6}$: [Limitations And Societal Impact] The limitations are not adequately mentioned, potential negative societal impact of their work are included in the manuscript.
>
> $\textbf{A6}$: Our work deals with synthetic experiments on image-based datasets. Therefore we believe there would not be negative social impact of our work. Nevertheless, following the suggestion of Comment \#4 from Reviewer 2, we would extend the discussion on the limitation of our work by thoroughly analyzing the potential cases that our method could or could not effectively tackle.

---

> > ### Comment · Reviewer_Ng2E · 2021-08-31
> > **Response to Authors**
> >
> > Thanks for providing the clarifications and new experiments. A minor comment is attached below:
> >
> > * "Both works have novel motivations to tackle the problems (i.e., EBM and neuromodulation) though their empirical results are a bit weak."
> >
> > The EBM work performs experiments in the Class-IL setting and is shown to outperform memory-based approaches. The neuromodulation work also outperforms other in Task-IL and Class-IL settings. The results are not directly comparable to NCCL to difference in the hyper parameters but it is key to note that non-memory-buffer-based approaches can have low memory an computational complexity while achieving accuracy similar to memory-buffer based approaches.

---

> > > ### Author Response · Authors · 2021-08-31
> > > **Response to Reviewer Ng2E**
> > >
> > > We sincerely thank the reviewer for sharing the in-depth thoughts on the non-memory-buffer-based approaches.
> > >
> > >
> > > We have checked out the two works again and we agree that both works contribute promising new directions to solve the continual learning problem and their performance is impressive under specified continual learning settings. We consider extending our task-incremental model to deal with class-incremental problems as a promising future research direction where [6] would serve as an crucial baseline. We look forward to the publication of both papers and the official release of implementation for [3] so that we could have a fair comparison with it under valid problem settings.
> > >
> > >
> > > We thank the reviewer again for extending the discussion before the end of the review period.

---

> > > > ### Comment · Reviewer_Ng2E · 2021-09-10
> > > > **Response to Authors**
> > > >
> > > > Thanks for acknowledging the differences in the task-and class-incremental learning modalities. This should be discussed in the revised version of the paper for clarity. I am updating my score since the authors addressed most of my concerns. However, having the class-increment learning results and comparison with existing literature (for example by re-using open source implementations such as: https://github.com/aimagelab/mammoth) would have provided more insights into the strengths and weakness of the proposed approach.

---

### Official Review · Reviewer_Eiq8 · 2021-07-18

**Rating:** 7
**Confidence:** 4

**Summary:**

The paper proposed a new online continual learning approach using neuron calibration techniques including model weight calibration and feature calibration. These calibration procedures only increase about 1x% model size which is reasonable. The performance on several datasets is impressive compared to other baselines. One surprising result is that the proposed method can even outperform offline learning (Fig. 2).

**Ethical Concerns:**

No.

**Limitations And Societal Impact:**

It seems to online continual learning is still a very new task to be studied. Hence, experimental settings are not aligned across papers.
Other than comparing with [21], why not adapting other settings or what's the issue of other settings?

**Main Review:**

Originality:
The proposed task agnostic calibration techniques are novel for online continual learning.
These calibration procedures only increase about 1x% model size which is reasonable.

Quality:
The results on many datasets are impressive, especially on the mini-ImageNet dataset.
Missing reference or experiment
1. [32] is indeed very similar to the proposed method excepted the method is not originally designed for online continual learning. However, I wonder what its performance is, if once naively apply the method on the online continual learning dataset.
2. A few latest works are missing in the discussion:
* Mitigating Forgetting in Online Continual Learning via Instance-Aware Parameterization. Chen et al. NeurIP'20.
* Continual Learning with Node-Importance-based Adaptive Group Sparse Regularization. Jung et al. NeurIP'20.

Significance:
The most impressive finding is in Fig. 2 where the proposed method can even outperform offline learning.
Rather than referring to [21], it will be even more impressive if some diagnoses can be done to understand the reason better.

Clarity:
The paper is well written with clear messages, equations, and figures.

**Time Spent Reviewing:**

3

---

> ### Author Response · Authors · 2021-08-10
> **Responses to Comments from Reviewer Eiq8**
>
> We thank the reviewer for the encouraging comments and constructive suggestions to improve our work.
>
> $\textbf{Q1}$: [32] is indeed very similar to the proposed method excepted the method is not originally designed for online continual learning. However, I wonder what its performance is, if once naively apply the method on the online continual learning dataset.
>
>
> $\textbf{A1}$:  We thank the reviewer for noticing [32] which has been an inspiring work to ours. We would like to first succinctly introduce the method. [32] slightly alters the original model architecture by applying additional calibrator parameters over a backbone model. Their calibrator is task-specific and trained on the second task onward. The backbone is shared across all the tasks and it is trained only on the first task and kept frozen thereafter. Thus the method comes with a strong assumption that the backbone parameter learned by the first task could be effective in representing the data from all its following tasks.
>
>
> The comparison result between our method and [32] is available in Table A5 from the supplementary material. The original version of [32] performs not very promising so we adapted the model and achieved much better performance by updating both the backbone and calibrator parameters from the tasks after the first one. Though the adapted model become comparable to our method, it maintains a task-specific set of parameters and such parameter isolation could naturally result in better performance against catastrophic forgetting compared to the attempts of using a unified model to tackle all the problems. The results show that our method leads to promising results to tackle OCL problems when using a unified model.
>
> $\textbf{Q2}$: A few latest works are missing in the discussion: mitigating forgetting and active group sparse regularization.
>
> $\textbf{A2}$: Thanks for referring to the related works. The two works tackle catastrophic forgetting in CL by learning optimal model connective path with reinforcement learning and group sparsity regularization. We will add detailed discussion with those works in our paper.
>
>
> $\textbf{Q3}$: The most impressive finding is in Fig. 2 where the proposed method can even outperform offline learning.Rather than referring to [21], it will be even more impressive if some diagnoses can be done to understand the reason better.
>
> $\textbf{A3}$: The observation that with large replay memory capacity, the performance of online continual learning method could outperform *offline* aligns with the part of the results reported in [21]. The reason could be that experience replay-based continual learning method poses an effective curriculum to learn multiple tasks from the interplay of tasks. Learning the tasks continuously could offer a positive forward transfer to the tasks learned at later stage while replaying the old tasks with expressive replay memory could potentially result in a positive backward transfer effect so that the performance of new task and old tasks could potentially increase concurrently. This observation is very inspiring to motivate future research on (O)CL as (O)CL approaches could potentially outperform the performance of multitask learning.
>
>
> $\textbf{Q4}$: [Limitations And Social Impact] Online continual learning is still a very new task to be studied. Hence, experimental settings are not aligned across papers. Why not adapting other settings or what's the issue of other settings other than comparing with [21]?
>
> $\textbf{A4}$: We thank the reviewer for raising up this important issue. Overall the main settings for OCL problems consists of the following three perspectives: dataset (including how to form tasks), model architecture and replay buffer size. Most of the disagreement on task setting among recent works is in terms of the replay buffer size. Apart from [21], earlier works adopt replay buffer of 200, 500 or greater than that (each task). In our work, we present primary results with a replay memory size of 50, but in addition we also provide results under the values of 100,150, 200, 250. Therefore, we think the experimental setting we adopt is reasonable for OCL problems. We will provide detailed discussion on the problem setting  in the [Limitations and Social Impact] section and call for more attention on the alignment of problem setting issue.

---

> > ### Comment · Reviewer_Eiq8 · 2021-09-04
> > **Responses to author.**
> >
> > Thanks for addressing my concerns and questions.
> > I am happy with the rebuttal and hope that the evaluation of OCL can converge to a standard benchmark in the near future.

---

### Official Review · Reviewer_zHgC · 2021-07-19

**Rating:** 6
**Confidence:** 3

**Summary:**

This paper presents a rehearsal based calibration technique for continual learning. The method learns parameters used to scale the parameters of every layer in the network, as well as parameters to scale the output of every layer. The network is trained in an interleaved fashion: first the scaling parameters are held fixed and the base network is trained on batches from the current task and examples from rehearsal memory. Then the base network is held fixed and the scaling parameters are trained using examples just from the rehearsal memory. This approach is tested on several different benchmarks and shows strong results.

**Limitations And Societal Impact:**

The authors have adequately addressed this.

**Main Review:**

The paper shows strong results on several continual learning benchmarks. It introduces a novel calibration technique that expands work in the line regularization based approaches for continual learning. Despite good results, I have a few issues/questions:

- Given that you are using a rehearsal based approach, you should have included the basic [Tiny Episodic Memory baseline](https://arxiv.org/pdf/1902.10486.pdf)
- I would have liked to see an ablation where you used only one of either parameter calibration or feature calibration, and turned the other off (instead of replacing with dropout), to see whether both are actually necessary.
- Several of the baseline methods listed (GEM, AGEM) use a task id during inference to specify the output labels. The supplementary lists them, as well as your approach, as not using task ids. Does your approach use task ids during training/inference to specify the valid output space?
- The description of $\mathcal{L_c}$ is a bit confusing, and several things are defined significantly after they are used ($S(), \hat{z}, \hat{z}_k$). The description of the Fisher information matrix, what it represents/how its being used can also be improved.
- There is a "stabilization issue" mentioned on page 5, but it is a bit unclear what that issue refers to, and how term b in the loss fixes it.
- For your baselines: does offline simply refer to a multitask baseline? Is there a limit on how many epochs you train the offline for?
- On page 8, you mention that the learning accuracy of your method is better than or equal to the offline method for all datasets. Did you mean to say independent, since offline does not have a learning accuracy reported? Also, a potentially more useful baseline comparison than independent (only for learning accuracy) would be finetune, as independent might be handicapped since it has to start from random initialization for every task.
- The checklist mentions you have provided code, but I could not find it. The github link in the supplementary had an empty repo. Could you please update the code?

Edit after discussion: I am raising my score to a 6.

**Time Spent Reviewing:**

5

---

> ### Author Response · Authors · 2021-08-10
> **Responses to Comments from Reviewer zHgC**
>
> We thank the reviewer for the insightful comments and suggestions to improve our work.
>
>
> $\textbf{Q1}$: Given that you are using a rehearsal based approach, you should have included the basic
> *Tiny Episodic Memory baseline*.
>
> $\textbf{A1}$:  We thank the reviewer for suggesting this baseline (we denote it as TinyER). TinyER promotes experience rehearsal based approach for continual learning. Its key focus is how to select the experience to be stored in the memory, which often requests the access to the entire samples for each task to evaluate their importance for the selection. Therefore such methods suggested in TinyER are not applicable to the online continual learning setting where the samples are observed as a one-pass stream. Nevertheless we will extend the discussion on TinyER in our related work.
>
>
> $\textbf{Q2}$: Does your approach use task ids during training/inference to specify the valid output space?
>
> $\textbf{A2}$: Our approach uses task ids during training/inference.  All the methods considered in our work tackle a task-incremental continual learning setting in which task ids are used during training/inference. We would like to clarify that the *task-agnostic* property mentioned in the paper actually refers to the case that our model architecture is task-agnostic, i.e., do not need task-id to query task-specific sub-model while many baselines such as *independent* and *CTN* would require task-id. We will carefully revise the writing in the paper and appendix to avoid misunderstanding.
>
> $\textbf{Q3}$: The description of $\mathcal{L}_c$ is a bit confusing, and several things are defined significantly after they are used $S(), \hat{z}, \hat{z}_k$.
>
> $\textbf{A3}$: Thanks for the reviewer's comment on the definition of the terms. We will remove the definitions from 204-205 into their previous paragraph.
>
> $\textbf{Q4}$:  There is a "stabilization issue" mentioned on page 5, but it is a bit unclear what that issue refers to, and how term $b$ in the loss fixes it.
>
> $\textbf{A4}$: Stabilization issue is related to the *stability-plasticity* dilemma for CL we introduced in Lines 39-44. In loss $\mathcal{L}_c$, we use term (a)  to enforce stability at the parameter-level and term(b) to enforce stability at the logit output-level.
>
> $\textbf{Q5}$: Does *offline* simply refer to a multitask baseline? Is there a limit on how many epochs you train the *offline* for?
>
> $\textbf{A5}$: *Offline* refers to a method with a different data observation regime than online CL where all the data for all the tasks are available at all time. Offline is similar with the offline multitask learning setting.  We do not specify a limit on the number of epochs for training.
>
> $\textbf{Q6}$: On page 8, you mention that the learning accuracy of your method is better than or equal to the offline method for all datasets. Did you mean to say *independent*, since offline does not have a learning accuracy reported?
>
> $\textbf{A6}$: We thank the reviewer for enquiring on it. Yes, the method mentioned on Line 291 of Page 8 should refer to the *independent* baseline. We will carefully revise the paper to avoid any further misunderstanding.
>
> $\textbf{Q7}$: The checklist mentions you have provided code, but the github link in the supplementary had an empty repo.
>
> $\textbf{A7}$: We spent time on cleaning the code and generate detailed documentations for the various experiments. We will release the code for our method and all the baselines for each experiment after the paper is accepted.

---

> > ### Comment · Reviewer_zHgC · 2021-08-31
> > **Follow up**
> >
> > Hi,
> >
> > Sorry for the late response, a couple of follow up comments.
> > 1. I think the Tiny episodic memory paper actually does all of it's experiments in an online setting. All of the strategies it uses to select examples are fairly simple, online approaches that only use the batch that is currently being processed and the episodic memory, and it outperforms methods like GEM, AGEM, and MER.
> > 2. Do you have an experiment/ablation where you use only one of feature/parameter calibration? Specifically, I was hoping to see how they individually affect performance compared to the baseline without adding the extra dropout noise in the ablations presented in the paper.
> >
> > Thanks

---

> > > ### Author Response · Authors · 2021-09-03
> > > **Reply to Reviewer zHgC**
> > >
> > > We thank Reviewer zHgC for extending the discussion about our paper and raising up the two related follow-up questions.
> > > ,
> > >
> > >
> > > 1. The tiny episodic memory paper (denoted as Tiny-EM hereafter)
> > >
> > > The reviewer is right that Tiny-EM could deal with online CL problems and its data selection strategy works on per mini-batch level. Since Tiny-EM focuses on evaluating the online CL models with restricted memory, i.e., adopting a setting of 1/3/5/15 samples per class to construct the episodic memory, we conducted additional experiments comparing our method NCCL with the various experience replay methods presented in Tiny-ER under its specified memory settings. The results on Permuted MNIST dataset and Split CIFAR-100 dataset is shown in Table 1. Because we tried various ways but could not reproduce the results shown in Tiny-EM paper and there are several other open issues in Tiny-ER repo reporting the same case of failure in reproducing the result, we show the original scores presented in the paper for the ER baselines.
> > >
> > > ,
> > >
> > >
> > >
> > > Table 1. ACC (Averaged Accuracy) on $\textbf{ Permuted MNIST}$ dataset (col: number of samples per class; row: result for a method)
> > >
> > > samples / class    |        1            |         3       |        5       |        15       |
> > >
> > > ER-RINGBUFFER   |    70.2 ($\pm$ 0.56)  |   73.5 ($\pm$ 0.43)   |    75.8 ($\pm$ 0.24)   |    79.4 ($\pm$ 0.43)
> > >
> > > ER-MOF,,,,,,,,,,,,,,,     |    69.9 ($\pm$ 0.68)  |   74.9 ($\pm$ 0.49)   |    78.3 ($\pm$ 0.19)   |    81.2 ($\pm$ 0.28)
> > >
> > > EF-K-MEANS,,,,,,,,    |    70.5 ($\pm$ 0.42)  |   74.7 ($\pm$ 0.62)   |    76.7 ($\pm$ 0.51)   |    79.1 ($\pm$ 0.23)
> > >
> > > ER-RESERVOIR ,,,   |    68.9 ($\pm$ 0.89  |   75.2 ($\pm$ 0.38)   |    76.2 ($\pm$ 0.38)   |    79.8 ($\pm$ 0.26)
> > >
> > > NCCL ,,,,,,,,,,,,,,,  ,,,,       |    72.9 ($\pm$ 0.69)  |   79.8  ($\pm$ 0.52)   |    84.6 ($\pm$ 0.42)   |    88.2 ($\pm$ 0.37)
> > >
> > >
> > >
> > > ,
> > >
> > >
> > >
> > > Table 2. ACC (Averaged Accuracy) on $\textbf{Split CIFAR}$ dataset (col: number of samples per class; row: result for a method)
> > >
> > >  samples / class,,     |        1            |         3       |        5       |        13       |
> > >
> > > ER-RINGBUFFER   |    56.2 ($\pm$ 1.93)  |   60.9 ($\pm$ 1.44)   |    62.6 ($\pm$ 1.77)   |    64.3 ($\pm$ 1.84)
> > >
> > > ER-MOF,,,,,,,,,,,,,,,     |    56.6 ($\pm$ 2.09)  |   59.9 ($\pm$ 1.25)   |    61.1 ($\pm$ 1.62)   |   62.7 ($\pm$ 0.63)
> > >
> > > EF-K-MEANS,,,,,,,,    |    56.6 ($\pm$ 1.40)  |   60.1 ($\pm$ 1.41)   |    62.2 ($\pm$ 1.20)   |   65.2 ($\pm$ 1.81)
> > >
> > > ER-RESERVOIR ,,,   |    53.1 ($\pm$ 2.66)  |   59.7 ($\pm$ 3.87)   |    65.5 ($\pm$ 1.99)   |    68.5 ($\pm$ 0.65)
> > >
> > > NCCL ,,,,,,,,,,,,,,,  ,,,,       |    42.7 ($\pm$ 1.69)  |   62.2  ($\pm$ 1.09)   |   69.1  ($\pm$ 0.42)   |    76.2 ($\pm$ 0.37)
> > >
> > > ,
> > >
> > > Overall, the results in Table 1 and Table 2 shows that except for 1 sample per class setting in Split CIFAR-100, our method could outperform the various ER baselines proposed in Tiny-EM with significant margins.
> > >
> > > ,
> > >
> > > We also wish to rectify that the results under the [ 5 samples per class] setting in Permuted MNIST for Tiny-EM (Table 3 in appendix) are derived under equivalent settings for the results we show in Table 2 of our paper, where we use a replay memory size of 50 which equals to 5 samples per class in MNIST. Similarly, the results under the [13 samples per class] setting in Split CIFAR-100 shown in Tiny-E< paper is equivalent to a replay memory size of 75 for us. Note that in Table 2 of our paper, even we use a smaller replay memory which have only 50 samples, our results  for Split CIFAR and Split MiniImagenet are much better than ER methods shown in Tiny-EM under the [13 samples per class] setting.
> > >
> > > ,
> > >
> > > ,
> > >
> > >
> > > 2. Additional ablation study for NCCL
> > >
> > > We have conducted the additional ablation study suggested by the reviewer. We denote the ablated model without feature calibration and weight calibration as NCCL(-f) and NCCL(-w), respectively. The results are shown in the tables below.
> > >
> > > ,
> > >
> > > Table 3. Ablation study results on $\textbf{Permuted MNIST}$.
> > >
> > >                           ACC (↑)        FM (↓)           LA (↑)
> > >       NCCL        83.47 ± 0.43      3.44 ± 0.26      86.59 ± 0.25
> > >       NCCL (-f)   82.98 ± 0.52      3.89 ± 0.65      86.38 ± 0.47
> > >       NCCL (-w)   81.72 ± 0.76      4.38 ± 0.77      85.72 ± 0.36
> > >
> > > ,
> > >
> > > Table 4. Ablation study results on $\textbf{Split CIFAR}$.
> > >
> > >                           ACC (↑)        FM (↓)           LA (↑)
> > >       NCCL        74.39 ± 1.01      4.88 ± 0.97      78.28 ± 0.53
> > >       NCCL (-f)   73.88 ± 0.96      5.62 ± 2.05      78.63 ± 0.97
> > >       NCCL (-w)   70.02 ± 0.71      3.46 ± 0.91      73.56 ± 1.18
> > > Overall, the effect of weight calibration is more significant than the feature calibration. Full NCCL with both calibration achieves best ACC.
> > >
> > >
> > > \,
> > >
> > > We thank the reviewer again for suggesting the additional related ablative experiments. We hope the additional experiments could help clarify the reviewer's concerns. We also welcome the reviewer(s) to raise further concerns if there is any.

---

### Decision · Program_Chairs · 2021-09-27

**Decision:**

Accept (Poster)

**Comment:**

This paper develops an approach to mitigate catastrophic forgetting in continual learning using neuron calibration techniques, including model weight and feature calibration, with rehearsal over the data. The resulting approach has impressive performance on task-incremental learning, especially compared to the other baselines. The paper is well-developed and clear with compelling results.

There are still a few remaining concerns that the authors are encouraged to address.  Most notably, adding class-incremental learning results and comparisons with related work would have provided more insights into the strengths and weakness of the proposed approach.  It would also help improve the strength of the results and broaden the paper. The authors are also encouraged to see if they can provide a deeper explanation of the results in Figure 2, helping to explain why the their method performs so well in comparison to offline learning. The authors' responses during the discussions included a number of clarifications and detail that should be added to the paper, including a promised clarification on what exactly is meant by the task-agnostic aspects of the approach.